# Analytical Composition of Differential Privacy via the Edgeworth Accountant

## Abstract

Many modern machine learning algorithms are in the form of a composition of simple private algorithms; thus, an increasingly important problem is to efficiently compute the overall privacy loss under composition. In this paper, we introduce the Edgeworth Accountant, an analytical approach to composing differential privacy guarantees of private algorithms. The Edgeworth Accountant starts by losslessly tracking the privacy loss under composition using the $f$-differential privacy framework (Dong et al., 2022), which allows us to express the privacy guarantees using privacy-loss log-likelihood ratios (PLLRs). As the name suggests, this accountant next uses the Edgeworth expansion (Hall, 2013) to upper and lower bound the probability distribution of the sum of the PLLRs. Moreover, by relying on a technique for approximating complex distributions by simple ones, we demonstrate that the Edgeworth Accountant can be applied to composition of any noise-addition mechanism. Owing to certain appealing features of the Edgeworth expansion, the $(\varepsilon, \delta)$-differential privacy bounds offered by this accountant are non-asymptotic, with essentially no extra computational cost, as opposed to the prior approaches in Koskela et al. (2020); Gopi et al. (2021), in which the running times are increasing with the number of compositions. Finally, we show our upper and lower $(\varepsilon, \delta)$-differential privacy bounds are tight in certain regimes of training private deep learning models and federated analytics.

## 1 Introduction

Differential privacy (DP) provides a mathematically rigorous framework for analyzing and developing private algorithms working on datasets containing sensitive information about individuals (Dwork et al., 2006). This framework, however, is often faced with challenges when it comes to analyzing the privacy loss of complex algorithms such as privacy-preserving deep learning and federated analytics (Ramage & Mazzocchi, 2020; Wang et al., 2021), which are composed of simple private building blocks. Therefore, a central question in this active area is to understand how the overall privacy guarantees degrade from the repetition of simple algorithms applied to the same dataset.

Continued efforts to address this question have led to the development of relaxations of differential privacy and privacy analysis techniques (Dwork et al., 2010; Dwork & Rothblum, 2016; Bun et al., 2018; Bun & Steinke, 2016). A recent flurry of activity in this line of research was triggered by Abadi et al. (2016), which proposed a technique called moments accountant for providing upper bounds on the overall privacy loss of training private deep learning models over iterations. A shortcoming of moments accountant is that the privacy bounds are generally not tight, albeit computationally efficient. This is because this technique is enabled by Rényi DP in Mironov (2017) and its following works (Balle et al., 2018; Wang et al., 2019), whose privacy loss profile can be lossy for many mechanisms. Alternatively, another line of works directly compose $(\varepsilon, \delta)$-DP guarantees via numerical methods such as the fast Fourier transform (Koskela et al., 2020; Gopi et al., 2021). This approach can be computationally expensive, as the number of algorithms under composition is huge, which unfortunately is often the case for training deep neural networks.

Instead, this paper aims to develop computationally efficient lower and upper privacy bounds for the composition of private algorithms with finite-sample guarantees[1] by relying on a new privacy defini-

---

[1] Here, "sample" refers to the number of compositions of DP algorithms. From now on we use the term "finite-sample" to mean that the bound is non-asymptotic in the number of compositions.

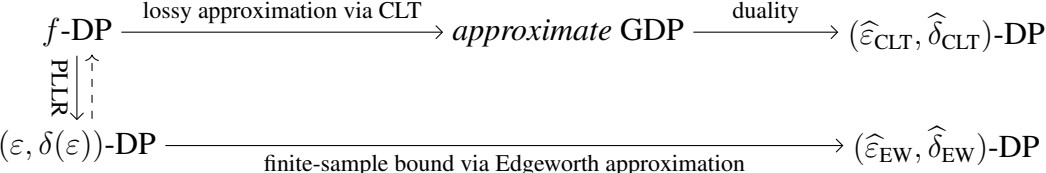

**Figure 1:** The comparison between the GDP approximation in Dong et al. (2022), and our Edgeworth Accountant. Both methods start from the exact composition using $f$-DP. Upper: Dong et al. (2022) uses a CLT type approximation to get a GDP approximation to the $f$-DP guarantee, then converts it to $(\varepsilon, \delta)$-DP via duality (Fact 1). Lower: We losslessly convert the $f$-DP guarantee to an exact $(\varepsilon, \delta(\varepsilon))$-DP guarantee, with $\delta(\varepsilon)$ defined with PLLRs in (3.1), and then take the Edgeworth approximation to numerically compute the $(\varepsilon, \delta)$-DP.

tion called $f$-differential privacy ($f$-DP (Dong et al., 2022)). $f$-DP offers a *complete* characterization of DP guarantees via a hypothesis testing interpretation, which was first introduced in Kairouz et al. (2015), and enables a precise tracking of the privacy loss under composition using a certain operation between the functional privacy parameters. Moreover, Dong et al. (2022) developed an approximation tool for evaluating the overall privacy guarantees using a central limit theorem (CLT), which can lead to *approximate* $(\varepsilon, \delta)$-DP guarantees using the duality between $(\varepsilon, \delta)$-DP and Gaussian Differential Privacy (GDP, a special type of $f$-DP) (Dong et al., 2022). While the $(\varepsilon, \delta)$-DP guarantees are asymptotically accurate, a usable finite-sample guarantee is lacking in the $f$-DP framework.

In this paper, we introduce the *Edgeworth Accountant* as an analytically efficient approach to obtaining finite-sample $(\varepsilon, \delta)$-DP guarantees by leveraging the $f$-DP framework. In short, the Edgeworth Accountant makes use of the Edgeworth approximation (Hall, 2013), which is a refinement to the CLT with a better convergence rate, to approximate the distribution of the sum of certain random variables that we refer to as privacy-loss log-likelihood ratios (PLLRs). By leveraging a Berry–Esseen type bound derived for the Edgeworth approximation, we obtain non-asymptotic upper and lower privacy bounds that are applicable to privacy-preserving deep learning and federated analytics. On a high level, we compare the approach of our Edgeworth Accountant to the Gaussian Differential Privacy approximation in Figure 1. Additionally, we note that while the rate of the Edgeworth approximation is well conceived, the explicit finite-sample error bounds are highly non-trivial. To the best of our knowledge, it is the first time such a bound has been established in the statistics and differential privacy communities and it is also of interest on its own.

We have made available two versions of our Edgeworth Accountant to better fulfill practical needs: the approximate Edgeworth Accountant (AEA), and the exact Edgeworth Accountant interval (EEAI). The AEA can give an estimate with *asymptotically* accurate bound for any number of composition $m$. By using higher-order Edgeworth expansion, such an estimate can be arbitrarily accurate, provided that the Edgeworth series converges, and therefore it is useful in practice to quickly estimate privacy parameters. As for the EEAI, it provides an accurate *finite-sample* bound for any $m$. It gives a rigorous bound on the privacy parameters efficiently.

Our proposal is very important as an efficiently computable DP-accountant. For the composition of $m$ identical mechanisms, our algorithm runs in $O(1)$ time to compute the privacy loss, and for the general case when we need to compose $m$ heterogeneous algorithms, the runtime becomes $O(m)$, which is information-theoretically optimal. In contrast, fast Fourier transform (FFT)-based algorithms (Gopi et al., 2021) provide accurate finite-sample bound, but can only achieve *polynomial* runtime for general composition of private algorithms. The suboptimal time-complexity of FFT-based methods leads to a large requirement of resources when $m$ is large, and a large $m$ is quite common in practice. For example, in deep learning and federated learning, $m$ is the number of iterations (rounds) and can be potentially very large. To make things worse, in real-world applications, the same dataset is often adaptively used or shared among different tasks. To faithfully account for the privacy loss, the DP accountant system has to track the cost of each iteration across different tasks, further increasing the number of compositions. Our EEAI serves as the first DP accountant method that simultaneously provides finite-sample guarantees, performs with optimal time complexity, and is very accurate (when $m$ is large), which can be a good supplement to the current toolbox.

The paper is organized as follows. We briefly summarize related work in privacy accounting of differentially private algorithms as well as our contributions in Section 1. In Section 2 we introduce the concept of $f$-DP and its important properties. We then introduce the notion of privacy-loss log-likelihood ratios in Section 3 and establish how to use them for privacy accountant based on distribution function approximation. In Section 4 we provide a new method, *Edgeworth Accountant*, that can efficiently and almost accurately evaluate the privacy guarantees, while providing finite-sample error bounds. Simulation results and conclusions can be found in Sections 5 and 6. Proofs and technical details are deferred to the appendices.

## 1.1 MOTIVATING APPLICATIONS

We now discuss two motivating applications: the NoisySGD (Song et al., 2013; Chaudhuri et al., 2011; Abadi et al., 2016; Bu et al., 2020) as well as the Federated Analytics and Federated Learning (Ramage & Mazzocchi, 2020; Wang et al., 2021). The analysis of DP guarantees of those applications are important yet especially challenging due to the *large* number of compositions involved. Our goal is primarily to devise a general tool to analyze the DP guarantees for these applications.

**NoisySGD.** NoisySGD is one of the most popular algorithms for training differentially private neural networks. In contrast to the standard SGD, the NoisySGD has two additional steps in each iteration: clipping (to bound the sensitivity of the gradients) and noise addition (to guarantee the privacy of models). The details of the NoisySGD algorithm is described in Algorithm 1 in Appendix B.

**Federated Analytics.** Federated analytics is a distributed analytical model, which performs statistical tasks through the interaction between a central server and local devices. To complete a global analytical task, in each iteration, the central server randomly selects a subset of devices to carry out local analytics and then aggregates results for the statistical analysis. The total number of iterations is usually very large[2] in federated analytics, requiring a tight analysis of its DP guarantee.

## 1.2 RELATED WORK

In this section, we present the following comparison of several existing works in DP accountant in Table 1. Specifically, we focus on their theoretical guarantees and the runtime complexity when the number of composition is $m$. We present a detailed survey of those DP accountants in Appendix A.

| Method | Finite-sample guarantee | Tightness of guarantee | Computational complexity |
|--------|-------------------------|------------------------|--------------------------|
| GDP/GDP-E | No | N/A | $O(1)$, $O(m)$ |
| MA | Only upper bound | Loose conversion to $(\varepsilon, \delta)$-DP | $O(1)$, $O(m)$ |
| FFT | Yes | Yes | $O(\sqrt{m})$, $O(m^{2.5})$ |
| **EA** | Yes | Yes* | $O(1)$, $O(m)$ |

**Table 1:** Comparison among different DP accountants. Each entry in the computation complexity contains two columns: (Left) the runtime for the composition of $m$ identical algorithms; (Right) the runtime for the composition of $m$ general algorithms. GDP: the Gaussian differential privacy accountant (Dong et al., 2022); GDP-E: the Edgeworth refinement to the GDP accountant (Zheng et al., 2020); MA: the moments accountant using Rényi-DP (Abadi et al., 2016); FFT: the fast Fourier transform accountant for privacy random variables (Gopi et al., 2021); **EA**: the Edgeworth Accountant we propose, including both the AEA (Definition 4.1), and the EEAI (Definition 4.2). *The guarantee of EA is tight when the order of the underlying Edgeworth expansion $k$ is high, or when $m$ is large for $k = 1$.

## 1.3 OUR CONTRIBUTIONS

We now briefly summarize our three main contributions.

**Improved time-complexity and estimation accuracy.** We propose a new DP accountant method, termed Edgeworth Accountant, which gives finite-sample error bound in constant/linear time complexity for the composition of identical/general mechanisms. In practice, our method outperforms GDP and moments accountant, with almost the same runtime.

---

[2]The number of iterations can be small for a single analytical task. However, in most practical cases, many statistical tasks are performed on the same base of users which leads to a large number of total iterations.

**A unified framework for efficient and computable evaluation of $f$-DP guarantee.** Though the evaluation of $f$-DP guarantee is #P-hard, we provide a general framework to efficiently approximate it. Leveraging this framework, any approximation scheme to the CDFs of the sum of privacy-loss log-likelihood ratios (PLLRs) can directly transform to a new DP accountant.

**Exact finite-sample Edgeworth bound analysis.** We are, to our best knowledge, the first to use Edgeworth expansion with finite-sample bounds in the statistics and machine learning communities. The analysis of the finite-sample bound of Edgeworth expansion is of its own interest, and has many potential applications. We further derive an explicit adaptive exponential decaying bound for the Edgeworth expansion of the PLLRs, which is the first such result for Edgeworth expansion.

## 2 PRELIMINARIES AND PROBLEM SETUP

In this section, we first define the notion of differential privacy and $f$-DP mathematically. We then set up the problem by revisiting our motivating applications.

A differentially private algorithm promises that an adversary with perfect information about the entire private dataset in use – except for a single individual – would find it hard to distinguish between its presence or absence based on the output of the algorithm (Dwork et al., 2006). Formally, for $\varepsilon > 0$, and $0 \leq \delta < 1$, we consider a (randomized) algorithm $M$ that takes as input a dataset.

**Definition 2.1.** A randomized algorithm $M$ is $(\varepsilon, \delta)$-DP if for any neighboring dataset $S, S'$ differing by an arbitrary sample, and for any event $E$, $\mathbb{P}[M(S) \in E] \leqslant \mathrm{e}^\varepsilon \cdot \mathbb{P}[M(S') \in E] + \delta$.

In Dong et al. (2022), the authors propose to use the trade-off between the type-I error and type-II error in place of a few privacy parameters in $(\varepsilon, \delta)$-DP. To formally define this new privacy notion, we denote by $P$ and $Q$ the distribution of $M(S)$ and $M(S')$, and let $\phi$ be a (possibly randomized) rejection rule for a hypothesis testing, where $H_0 : P$ vs. $H_1 : Q$. The trade-off function $f$ between $P$ and $Q$ is then defined as the mapping between type-I error to type-II error, that is, $f = T(P, Q) : \alpha \mapsto \inf_\phi \{1 - \mathbb{E}_Q[\phi] : \mathbb{E}_P[\phi] \leqslant \alpha\}$. This motivates the following definition.

**Definition 2.2.** A (randomized) algorithm $M$ is $f$-differentially private if $T(M(S), M(S')) \geqslant f$ for all neighboring datasets $S$ and $S'$.

The following facts about $f$-DP have been established in Bu et al. (2020); Dong et al. (2022).

**Fact 1** (Duality to $(\varepsilon, \delta)$-DP). A mechanism is $f$-DP if and only if it is $(\varepsilon, \delta(\varepsilon))$-DP for all $\varepsilon > 0$, with $\delta(\varepsilon) = 1 + f^*(-e^\varepsilon)$. Here $g^*(y) = \sup_{-\infty < x < \infty} yx - g(x)$ is the convex conjugate of $g$.

**Fact 2** (Composition). Letting $M_1$ and $M_2$ be two mechanisms, we define their composition algorithm $M$ as $M(S) = (M_1(S), M_2(S, M_1(S)))$. In general, the composition of more than two algorithms follows recursively. Given trade-off functions $f = T(P, Q)$ and $g = T(P', Q')$, let $f \otimes g = T(P \times P', Q \times Q')$. Assume $M_t$ is $f_t$-DP for $t = 1, \ldots, m$. The composition theorem states that their $m$-fold composition algorithm is $f_1 \otimes \cdots \otimes f_m$-DP, which is tight in general.

**Fact 3** (Subsampling). Consider the following two most common subsampling schemes: (1) (Poisson subsampling) for each individual in the dataset $S$, includes its datum in the subsample independently with probability $p$; (2) (Uniform subsampling) draws a subsample of $S$ that is chosen uniformly at random among all $s = |S|p$-sized subsets of $S$. Denote $\mathrm{Id}(\alpha) = 1 - \alpha$, and suppose an algorithm $M$ is $f$-DP. The subsampling theorem for $f$-DP states that the Poisson subsampled and uniform subsampled algorithm are both $\min\{f_p, f_p^{-1}\}^{**}$-DP, where $f_p = pf + (1-p)\,\mathrm{Id}$.

**Fact 4** (Gaussian Differential Privacy (GDP)). To deal with the composition of $f$-DP guarantees, Dong et al. (2022) introduce the concept of $\mu$-GDP, which is a special case of $f$-DP with $f = G_\mu = T(\mathcal{N}(0, 1), \mathcal{N}(\mu, 1))$. They prove that when all the $f$-DP guarantees are close to the identity, their composition is asymptotically a $\mu$-GDP with some computable $\mu$, which can then be converted to $(\varepsilon, \delta)$-DP via duality. However, it comes without a finite-sample bound.

With these facts, we can characterize the $f$-DP guarantee for motivating applications in Section 1.1.

**NoisySGD.** For a NoisySGD with $m$ iterations, subsampling ratio of $p$, and noise multiplier $\sigma$, it is $\min\{f, f^{-1}\}^{**}$-DP (Bu et al., 2020; Dong et al., 2022), with $f = \left(pG_{1/\sigma} + (1-p)\mathrm{Id}\right)^{\otimes m}$.

**Federated Analytics.** Suppose there are $m$ tasks, and each task is $f_i$-DP with $f_i = T(P_i, Q_i)$. Then the overall DP guarantee is characterized by $\bigotimes_{i=1}^m f_i$-DP.

It is easy to see that the $f$-DP guarantee of NoisySGD is a special case of the $f$-DP guarantee of federated analytics with each trade-off function being $f_i = \min\{f_p, f_p^{-1}\}^{**}$, that is, with identical composition of subsampled Gaussian mechanisms. Therefore, our goal is to efficiently and accurately evaluate the privacy guarantee of the general $\bigotimes_{i=1}^{m} f_i$-DP with an explicit finite-sample error bound.

## 3 PRIVACY-LOSS LOG-LIKELIHOOD RATIOS (PLLRS)

We aim to compute the explicit DP guarantees for general composition of trade-off functions of the form $f = \bigotimes_{i=1}^{m} f_i$. For the $i$-th composition, the trade-off function $f_i = T(P_i, Q_i)$ is realized by the two hypotheses: $H_{0,i} : w_i \sim P_i$ vs. $H_{1,i} : w_i \sim Q_i$, where $P_i, Q_i$ are two distributions. To evaluate the trade-off function $f = \bigotimes_{i=1}^{m} f_i$, we are essentially distinguishing between the two composite hypotheses $H_0 : \boldsymbol{w} \sim P_1 \times P_2 \times \cdots \times P_m$ vs. $H_1 : \boldsymbol{w} \sim Q_1 \times Q_2 \times \cdots \times Q_m$, where $\boldsymbol{w} = (w_1, ..., w_m)$ is the concatenation of all $w_i$'s. Motivated by the optimal test asserted by the Neyman-Pearson Lemma, we give the following definition.

**Definition 3.1.** The associated pair of *privacy-loss log-likelihood ratios (PLLRs)* is defined to be the logarithm of the Radon-Nikodym derivatives of two hypotheses under null and alternative hypothesis, respectively. Specifically, we can express PLLRs with respect to $H_{0,i}$ and $H_{1,i}$ as
$X_i \equiv \log\left(\frac{dQ_i(\xi_i)}{dP_i(\xi_i)}\right), Y_i \equiv \log\left(\frac{dQ_i(\zeta_i)}{dP_i(\zeta_i)}\right)$, where $\xi_i \sim P_i, \zeta_i \sim Q_i$. [3]

Note that the definition of PLLRs only depends on the two hypotheses. It allows us to convert the $f$-DP guarantee to a collection of $(\varepsilon, \delta)$-DP guarantees losslessly. The following proposition characterizes the relationship between $\varepsilon$ and $\delta$ in terms of the distribution functions of PLLRs.

**Proposition 3.2.** *Let $X_1, \ldots, X_m$ and $Y_1, \ldots, Y_m$ be the PLLRs defined above. Let $F_{X,m}, F_{Y,m}$ be the CDFs of $X_1 + \cdots + X_m$ and $Y_1 + \cdots + Y_m$, respectively. Then, the composed mechanism is $f$-DP (with $f = \bigotimes_{i=1}^{m} f_i$) if and only if it is $(\varepsilon, \delta)$-DP for all $\varepsilon > 0$ with $\delta$ defined by*

$$\delta = 1 - F_{Y,m}(\varepsilon) - e^{\varepsilon}(1 - F_{X,m}(\varepsilon)). \tag{3.1}$$

Proposition 3.2 establishes a relationship between $f$-DP and a collection of $(\varepsilon, \delta(\varepsilon))$-DP, which reflects the primal-dual relationship between them. Note that some special forms of this general proposition have been proved previously in terms of privacy loss random variables. See Balle & Wang (2018); Zhu et al. (2021); Gopi et al. (2021) for details. The key contribution of our Proposition 3.2 is that we express the $(\varepsilon, \delta)$-DP characterization of the #P-complete $f$-DP in terms of (3.1), which can be approximated directly. Of note, the above relationship is general in the sense that we make no assumption on the private mechanisms.

Definition 3.1 can be applied directly when $\frac{dQ_i(\xi_i)}{dP_i(\xi_i)}$ is easy to compute, which corresponds to the case without subsampling. To deal with the case with subsampling, one must take into account that the subsampled DP guarantee is the double conjugate of the minimum of two asymmetric trade-off functions (for example, recall the trade-off function of a single sub-sampled Gaussian mechanism is $\min\{f_p, f_p^{-1}\}^{**}$, where $f_p = (pG_{1/\sigma} + (1-p)\text{Id})$). In general, the composition of multiple subsampled mechanisms satisfies $f$-DP for $f = \min\{\otimes_{i=1}^{m} f_{i,p_i}, \otimes_{i=1}^{m} f_{i,p_i}^{-1}\}^{**}$. This general form makes the direct computation of the PLLRs through composite hypotheses infeasible, as it is hard to write $f$ as a trade-off function for some explicit pair of hypotheses $H_0$ and $H_1$. Therefore, instead of using one single sequence of PLLRs directly corresponding to $f$, we shall use a family of sequences of PLLRs. In general, suppose we have a mechanism characterized by some $f$-DP guarantee, where $f = \left(\inf_{\alpha \in \mathcal{I}}\{f^{(\alpha)}\}\right)^{**}$, for some index set $\mathcal{I}$. That is, $f$ is the tightest possible trade-off function satisfying all the $f^{(\alpha)}$-DP. Suppose further that for each $\alpha$, we can find a sequence of computable PLLRs corresponding to $f^{(\alpha)}$, which allows us to obtain a collection of $(\varepsilon, \delta^{(\alpha)}(\varepsilon))$-DP guarantees.

**Lemma 3.3.** *Suppose that for each $\alpha$, functions $f^{(\alpha)}$ and $\delta^{(\alpha)}$ satisfy that a mechanism is $f^{(\alpha)}$-DP if and only if it is $(\varepsilon, \delta^{(\alpha)}(\varepsilon))$-DP for all $\varepsilon > 0$. Then a mechanism is $f = \left(\inf_{\alpha \in \mathcal{I}}\{f^{(\alpha)}\}\right)^{**}$-DP if and only if it is $(\varepsilon, \sup_\alpha\{\delta^{(\alpha)}(\varepsilon)\})$-DP for all $\varepsilon > 0$.*

We defer the proof of this lemma to the appendices. The intuition is that both $\left(\inf_{\alpha \in \mathcal{I}}\{f^{(\alpha)}\}\right)^{**}$-DP and $(\varepsilon, \sup_\alpha\{\delta^{(\alpha)}(\varepsilon)\})$-DP correspond to the *tightest* possible DP-guarantee for the entire collection.

---

[3]For completeness, we explicitly require that all $\xi_i$ and $\zeta_i$ be independent.

Lemma 3.3 allows us to characterize the subsampled Gaussian mechanism using two sequences of PLLRs. As mentioned above, it is $f$-DP with $f = \min\{\otimes_{i=1}^m f_{i,p}, \otimes_{i=1}^m f_{i,p}^{-1}\}^{**}$, where each $f_{i,p} = (pG_{1/\sigma} + (1-p)\mathrm{Id})$. For the first part, the PLLRs corresponding to $\otimes_{i=1}^m f_{i,p} = (pG_{1/\sigma} + (1-p)\mathrm{Id})^{\otimes m}$ are given by $X_i^{(1)} = \log(1 - p + pe^{\mu\xi_i - \frac{1}{2}\mu^2})$, and $Y_i^{(1)} = \log(1 - p + pe^{\mu\zeta_i - \frac{1}{2}\mu^2})$, for $1 \leq i \leq m$, with $\xi_i \sim \mathcal{N}(0,1)$, $\zeta_i \sim p\mathcal{N}(0,1) + (1-p)\mathcal{N}(\mu,1)$. And for the second part, the PLLRs corresponding to $\otimes_{i=1}^m f_{i,p}^{-1} = ((pG_{1/\sigma} + (1-p)\mathrm{Id})^{-1})^{\otimes m}$ are given by $X_i^{(2)} = -\log(1 - p + pe^{\mu\zeta_i - \frac{1}{2}\mu^2})$, and $Y_i^{(2)} = -\log(1 - p + pe^{\mu\xi_i - \frac{1}{2}\mu^2})$, for $1 \leq i \leq m$, with $\xi_i \sim \mathcal{N}(0,1)$, $\zeta_i \sim p\mathcal{N}(0,1) + (1-p)\mathcal{N}(\mu,1)$. Now substituting $F_{X^{(1)},m}$ and $F_{Y^{(1)},m}$ by any approximation (for example, using the CLT or Edgeworth), we get a computable relationship in terms of the $(\varepsilon, \delta^{(1)}(\varepsilon))$-DP; and similarly, we can get a relationship in terms of the $(\varepsilon, \delta^{(2)}(\varepsilon))$-DP. We conclude that the subsampled Gaussian mechanism is $(\varepsilon, \max\{\delta^{(1)}(\varepsilon), \delta^{(2)}(\varepsilon)\})$-DP.

## 3.1 TRANSFERRED ERROR BOUND BASED ON CDF APPROXIMATIONS

As discussed above, Lemma 3.3 allows us to characterize the double conjugate of the infimum of a collection of $f^{(\alpha)}$-DPs via analyzing each sequence of PLLRs separately. As a result, our focus is to compute the bounds of $\delta^{(\alpha)}$ for each single trade-off function $f^{(\alpha)}$. To fulfill the purpose, we seek an efficient algorithm for approximating distribution functions of the sum of PLLRs, namely, $F_{X^{(\alpha)},m}, F_{Y^{(\alpha)},m}$. This perspective provides a general framework that naturally encompasses many existing methods, including fast Fourier transform (Gopi et al., 2021) and the characteristic function method (Zhu et al., 2021). They can be viewed as different methods for finding upper and lower bounds of $F_{X^{(\alpha)},m}, F_{Y^{(\alpha)},m}$. Specifically, we denote the upper and lower bounds of $F_{X^{(\alpha)},m}$ by $F_{X^{(\alpha)},m}^+$ and $F_{X^{(\alpha)},m}^-$, and similarly for $F_{Y^{(\alpha)},m}$. These bounds can be easily converted to the error bounds on privacy parameters of the form $g^{(\alpha)-}(\varepsilon) \leq \delta^{(\alpha)}(\varepsilon) \leq g^{(\alpha)+}(\varepsilon)$, for all $\varepsilon > 0$, where

$$
\begin{aligned}
g^{(\alpha)+}(\varepsilon) &= 1 - F_{Y^{(\alpha)},m}^-(\varepsilon) - e^{\varepsilon}(1 - F_{X^{(\alpha)},m}^+(\varepsilon)), \\
g^{(\alpha)-}(\varepsilon) &= 1 - F_{Y^{(\alpha)},m}^+(\varepsilon) - e^{\varepsilon}(1 - F_{X^{(\alpha)},m}^-(\varepsilon)).
\end{aligned}
\tag{3.2}
$$

Thus, the DP guarantee of $(\inf_\alpha f^{(\alpha)})^{**}$-DP in the form of $(\varepsilon, \delta(\varepsilon))$ satisfies $\sup_\alpha\{g^{(\alpha)-}(\varepsilon)\} \leq \delta(\varepsilon) \leq \sup_\alpha\{g^{(\alpha)+}(\varepsilon)\}$, for all $\varepsilon > 0$. To convert the guarantee of the form $(\varepsilon, \delta(\varepsilon))$ for all $\varepsilon > 0$ to the guarantee of the form $(\varepsilon(\delta), \delta)$ for all $\delta \in [0, 1)$, we can invert the above bounds on $\delta(\varepsilon)$ and obtain the bounds of the form $\varepsilon^-(\delta) \leq \varepsilon(\delta) \leq \varepsilon^+(\delta)$. Here $\varepsilon^+(\delta)$ is the largest root of equation $\delta = \sup_\alpha\{g^{(\alpha)+}(\cdot)\}$, and $\varepsilon^-(\delta)$ is the smallest non-negative root of equation $\delta = \sup_\alpha\{g^{(\alpha)-}(\cdot)\}$.

**Remark 3.4.** In practice, we often need to solve for those roots numerically, and we need to specify a finite range in which we find all the roots. In Appendix B, we exemplify how to find such range for NoisySGD, see Remark B.1 in the appendices for details.

## 4 EDGEWORTH ACCOUNTANT WITH FINITE-SAMPLE GUARANTEE

In what follows, we present a new approach, *Edgeworth Accountant*, based on the Edgeworth expansion to approximate the distribution functions of the sum of PLLRs. For simplicity, we demonstrate how to obtain the Edgeworth Accountant for any trade-off function $f^{(\alpha)}$ based on a single sequence of PLLRs $\{X_i^{(\alpha)}\}_{i=1}^m, \{Y_i^{(\alpha)}\}_{i=1}^m$. Henceforth, we drop the superscript $\alpha$ when it is clear from context. Specifically, we derive an approximate Edgeworth Accountant (AEA) and the associated exact Edgeworth Accountant interval (EEAI) for $f$ with PLLRs $\{X_i\}_{i=1}^m, \{Y_i\}_{i=1}^m$. We define AEA and EEAI for general trade-off function of the form $(\inf_\alpha f^{(\alpha)})^{**}$ in Appendix C.

### 4.1 EDGEWORTH ACCOUNTANT

To approximate the CDF of a random variable $X = \sum_{i=1}^m X_i$, we introduce Edgeworth expansion in its most general form, where $X_i$'s are independent but not necessarily identical. Such generality allows us to account for composition of heterogeneous DP algorithms. Suppose $\mathbb{E}[X_i] = \mu_i$ and $\gamma_{p,i} := \mathbb{E}[(X_i - \mu_i)^p] < +\infty$ for some $p \geq 4$. We define $B_m := \sqrt{\sum_{i=1}^m \mathbb{E}[(X_i - \mu_i)^2]}$, and $\sum_{i=1}^m \mu_i = M_m$. So, the standardized sum can be written as $S_m := (X - M_m)/B_m$. We

denote $E_{m,k,X}(x)$ to be the $k$-th order Edgeworth approximation of $S_m$. Note that the central limit theorem (CLT) can be viewed as the $0$-th order Edgeworth approximation. The first-order Edgeworth approximation is given by adding one extra order $O(1/\sqrt{m})$ term to the CLT, that is, $E_{m,1,X}(x) = \Phi(x) - \frac{\lambda_{3,m}}{6\sqrt{m}}\left(x^2 - 1\right)\phi(x)$. Here, $\Phi$ and $\phi$ are the CDF and PDF of a standard normal distribution, and $\lambda_{3,m}$ is a constant to be defined in Lemma 4.3. It is known that (see for example, Hall (2013)) the Edgeworth approximation of order $p$ has an error rate of $O(m^{-(p+1)/2})$. This desirable property motivates us to use the rescaled Edgeworth approximation $G_{m,k,X}(x) = E_{m,k,X}\left((x - M_m)/B_m\right)$ and $G_{m,k,Y}(x) = E_{m,k,Y}\left((x - M_m)/B_m\right)$ to approximate $F_{X,m}(x)$ and $F_{Y,m}(x)$, respectively, in (3.1). This is what we term the *approximate Edgeworth Accountant* (AEA).

**Definition 4.1** (AEA). The $k$-th order AEA that defines $\delta(\varepsilon)$ for $\varepsilon > 0$ is given by $\delta(\varepsilon) = 1 - G_{m,k,Y}(\varepsilon) - e^\varepsilon(1 - G_{m,k,X}(\varepsilon))$, for all $\varepsilon > 0$.

Asymptotically, AEA is an exact accountant, due to the rate of convergence Edgeworth approximation admits. In practice, however, the finite-sample guarantee is still missing since the exact constant of such rate is unknown. To obtain a computable $(\varepsilon, \delta(\varepsilon))$-DP bound via (3.1), we require the finite-sample bounds on the approximation error of the CDF for any finite number of iterations $m$. Suppose that we can provide a finite-sample bound using Edgeworth approximation of the form $|F_{X,m}(x) - G_{m,k,X}(x)| \leq \Delta_{m,k,X}(x)$, where $\Delta_{m,k,X}(x)$ is computable. Then we have

$$F^+_{X,m}(x) = G_{m,k,X}(x) + \Delta_{m,k,X}(x) \quad \text{and} \quad F^-_{X,m}(x) = G_{m,k,X}(x) - \Delta_{m,k,X}(x), \tag{4.1}$$

and similarly for $F_{Y,m}$. We now define the *exact Edgeworth Accountant interval* (EEAI).

**Definition 4.2** (EEAI). The $k$-th order EEAI associated with privacy parameter $\delta(\varepsilon)$ for $\varepsilon > 0$ is given by $[\delta^-, \delta^+]$, where for all $\varepsilon > 0$

$$\begin{aligned}
\delta^-(\varepsilon) &\equiv 1 - G_{m,k,Y}(\varepsilon) - \Delta_{m,k,Y}(\varepsilon) - e^\varepsilon(1 - G_{m,k,X}(\varepsilon) + \Delta_{m,k,X}(\varepsilon)), \\
\delta^+(\varepsilon) &\equiv 1 - G_{m,k,Y}(\varepsilon) + \Delta_{m,k,Y}(\varepsilon) - e^\varepsilon(1 - G_{m,k,X}(\varepsilon) - \Delta_{m,k,X}(\varepsilon)).
\end{aligned} \tag{4.2}$$

To bound the EEAI, it suffices to have a finite-sample bound on $\Delta_{m,k,X}(\varepsilon)$ and $\Delta_{m,k,Y}(\varepsilon)$.

### 4.2 UNIFORM BOUND ON PLLRS

We now deal with the bound of the Edgeworth approximation on PLLRs in (4.1). Our starting point is a uniform bound of the form $\Delta_{m,k,X}(x) \leq c_{m,k,X}$, for all $x$. The bound for $\Delta_{m,k,Y}(x)$ follows identically. To achieve this goal, we follow the analysis on the finite-sample bound in Derumigny et al. (2021). We state the bound of the first-order Edgeworth expansion.

**Lemma 4.3.** *Define the average individual standard deviation $\bar{B}_m := B_m/\sqrt{m}$ and the average standardized $r$-th cumulant as $\lambda_{k,m} := \frac{1}{m}\sum_{j=1}^m k_{r,j}/\bar{B}_m^3$, where $k_{r,j}$ is the $r$-th centralized cumulant of the $j$-th sample. With bounded moments of order four, that is, $\gamma_{4,i} < +\infty$ for $1 \leq i \leq m$, we have the (uniform) bound on Edgeworth expansion as*

$$\Delta_{m,1,X} \leq \frac{0.1995\widetilde{K}_{3,m}}{\sqrt{m}} + \frac{0.031\widetilde{K}_{3,m}^2 + 0.195K_{4,m} + 0.054|\lambda_{3,m}|\widetilde{K}_{3,m} + 0.038\lambda_{3,m}^2}{m} + r_{1,m},$$

*where $K_{p,m} := m^{-1}\sum_{i=1}^m \mathbb{E}\left[|X_i - \mu_i|^p\right]/\left(\bar{B}_m\right)^p$, which is the average standardized $p$-th absolute moment, and $\widetilde{K}_{3,m} := K_{3,m} + \frac{1}{m}\sum_{i=1}^m \mathbb{E}|X_i - \mu_i|\gamma_{2,i}/\bar{B}_m^3$. Here $r_{1,m}$ is a remainder term of order $O(1/m^{5/4})$ that depends only on $K_{3,m}, K_{4,m}$ and $\lambda_{3,m}$, and is defined in Equation (H.1).*

Note that this lemma deals with the first-order Edgeworth approximation which can be generalized to the higher-order Edgeworth approximations. We present the analysis of the second- and third-order in the appendices. The expression of $r_{1,m}$ only involves real integration with known constants which can be numerically evaluated in constant time.

**Remark 4.4.** The precision of the EEAI highly depends on the rate of the finite-sample bound of the Edgeworth expansion. Any better bounds for higher-order Edgeworth expansions can be directly applied to our EEAI by substituting $\Delta_{m,k,X}(\varepsilon)$, here we simply demonstrate when $k = 1$ leveraging the first-order expansion. Observe that Lemma 4.3 gives a bound of order $O(1/\sqrt{m})$ due to the reason that we want to deal with general independent but not necessarily identical random variables. We demonstrate how one can obtain a $O(1/m)$ rate in the i.i.d. case in Appendix H. Our current first-order bound is primarily useful *when $m$ is large enough*, but a bound for higher-order Edgeworth expansions can further improve the precision for all $m$.

### 4.3 ADAPTIVE EXPONENTIAL DECAYING BOUND FOR NOISYSGD

One specific concern of the bound derived in the previous section is that it is uniform in $\varepsilon$. Note that in (3.1), there is an amplification factor of error by $e^\varepsilon$ in front of $F_{X,m}$. Therefore, as long as $\varepsilon$ grows in $m$ with order at least $\varepsilon \sim \Omega(\log m)$, the error term in (3.1) scales with order $e^{\Omega(\log m)}/O(m) = \Omega(1)$.

In this section, we study the compositions of subsampled Gaussian mechanism (including NoisySGD and many federated learning algorithms), where we are able to improve the previous bound when $\varepsilon$ is large. Informally, omitting the dependence on $m$, we want to have a bound of the form $|F_{X,m}(\varepsilon) - G_{m,k,X}(\varepsilon)| = O(e^{-\varepsilon^2})$ to offset the effect of $e^\varepsilon$ in front of $F_{X,m}$. To this end, we first prove that the tail bound of $F_{X,m}(\varepsilon)$ is of order $O(e^{-\varepsilon^2})$, with exact constant. Combining with the tail behavior of the Edgeworth expansion, we conclude that the difference has the desired convergence rate. Following the discussion in Section 3, we need to calculate the bounds for two sequences of PLLRs separately. Here we focus on the sequence of PLLRs corresponding to $\left(pG_{1/\sigma} + (1-p)\text{Id}\right)^{\otimes m}$. These PLLRs are given by $X_i = \log(1 - p + pe^{\mu\xi_i - \frac{1}{2}\mu^2})$, where $\xi_i \sim N(0,1)$. The following theorem characterizes the tail behavior of $F_{X,m}$. The tail bound of the sum of the other sequence of PLLRs corresponding to $((pG_{1/\sigma} + (1-p)\text{Id})^{-1})^{\otimes m}$ has the same rate, and can be proved similarly.

**Theorem 1.** *There exist some positive constant $a$, and some associated constant $\eta(a) > 0$, such that the tail of $F_{X,m}$ can be bounded as $1 - F_{X,m}(\varepsilon) = \mathbb{P}\left(\sum_{i=1}^m X_i \geq \varepsilon\right) \leq 2\exp\left(-\frac{(\varepsilon+m\eta)^2}{8m\tau^2}\right)$, where*

$$\tau^2 = \max\left\{\frac{(\log(1-p+pe^{\mu a - \frac{1}{2}\mu^2}) + \mu(a^+ - a) - \log(1-p))^2}{4}, \mu^2, \frac{(a^+ - a)^2\mu^2}{2\log(\Phi(a^+) - \Phi(a))}\right\} \text{ and } a^+ = \frac{\phi(a)}{1 - \Phi(a)}.$$

The proof of Theorem 1 is deferred to Appendix G along with its dependent technical lemmas. From the above theorem, we know that the tail of $F_{X,m}(\varepsilon)$ is $O(e^{-\max\{\varepsilon^2/m, m\}}) = o(e^{-\varepsilon})$, as long as $\varepsilon = o(m)$. Note that in this case, the tail of the rescaled Edgeworth expansion is of the same order $O(e^{-\max\{\varepsilon^2/m, m\}}) = o(e^{-\varepsilon})$. Therefore, we can give a finite-sample bound of the same rate for the difference between $F_{X,m}(\varepsilon)$ and its approximation $G_{m,k,X}$ at large $\varepsilon$. Note that this finite-sample bound scales better than uniform bound in Lemma 4.3 when $m$ and $\varepsilon$ are large.

### 4.4 EXTENSION TO OTHER MECHANISMS

Note that our analysis framework is applicable to a wide range of common noise-adding mechanisms. Specifically, Lemma 4.3 only requires the distribution of PLLRs to have bounded fourth moments. And for many common mechanisms, a counterpart of Theorem 1 can be proved similarly. We now demonstrate how to generalize our analysis to the Laplace Mechanism.

**The Laplace Mechanism.** It is straightforward to verify that the trade-off function for subsampled Laplace Mechanisms is given by $\min\{(pL_\mu + (1-p)\text{Id})^{\otimes m}, ((pL_\mu + (1-p)\text{Id})^{-1})^{\otimes m}\}^{**}$, where $L_\mu = T(\text{Lap}(0,1), \text{Lap}(\mu,1))$. The two associated sequences of PLLRs $X_i$ and $Y_i$ can be expressed as: $X_i^{(1)} \equiv \log\left(1 - p + pe^{|\xi| - |\xi - \mu|}\right), Y_i^{(1)} \equiv \log\left(1 - p + pe^{|\zeta| - |\zeta - \mu|}\right)$, and $X_i^{(2)} \equiv -\log\left(1 - p + pe^{|\zeta| - |\zeta - \mu|}\right), Y_i^{(2)} \equiv -\log\left(1 - p + pe^{|\xi| - |\xi - \mu|}\right)$, where $\xi \sim \text{Lap}(0,1), \zeta \sim p\text{Lap}(\mu,1) + (1-p)\text{Lap}(0,1)$. Note that all the PLLRs are bounded and thus sub-Gaussian. This implies that we can apply Lemma 4.3 directly and also bound the tail similar to Theorem 1.

**Proposition 4.5.** *Denote $\eta = -\max\left\{\mathbb{E}(X_i^{(1)}), \mathbb{E}(X_i^{(2)})\right\} > 0$. The tail of the sum of both sequence of PLLRs under the Laplace Mechanism has the following inverse exponential behavior, $\max\left\{\mathbb{P}\left(\sum_{i=1}^m X_i^{(1)} \geq \varepsilon\right), \mathbb{P}\left(\sum_{i=1}^m X_i^{(2)} \geq \varepsilon\right)\right\} \leq \exp\left(-\frac{2(\varepsilon+m\eta)^2}{m\tau^2}\right)$, where $\tau^2 = (\log(1 - p + pe^\mu) - \log(1 - p + pe^{-\mu}))^2$.*

## 5 NUMERICAL EXPERIMENTS

In this section, we illustrate the advantages of Edgeworth Accountant by presenting the plots of DP accountant curves under different settings. Specifically, we plot the privacy curve of $\varepsilon$ against number of compositions and compare our methods (AEA and EEAI) with existing DP accountants. We provide the implementation of our Edgeworth Accountant in Appendix D.

**The AEA.** We first demonstrate that our proposed approximate Edgeworth Accountant (AEA) is indeed very accurate, outperforming the existing Rényi DP and the CLT approximations in experiments. The first experiment has the same setting as in Figure 1(b) in Gopi et al. (2021), where the authors report that both RDP and GDP are inaccurate, whereas the second setting corresponds to a real federated learning task. The results are shown in Figure 2, where we describe the specific settings in the caption. For each sub-figure, the dotted lines "FFT_LOW" and "FFT_UPP" denote the lower and upper bound computed by FFT (Gopi et al., 2021) which are used as the underlying ground truth. The "GDP" curve is computed by the CLT approximation (Bu et al., 2020), the "RDP" curve is computed by moments accountant using Rényi DP with subsampling amplification (Wang et al., 2019), and the "EW_EST" curve is computed by our (second-order) AEA. As is evident from the figures, our AEA outperforms the GDP and RDP.

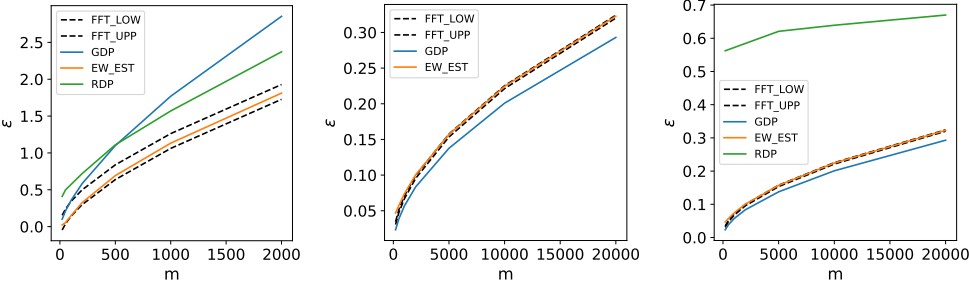

**Figure 2:** The privacy curve computed via several different accountants. Left: The setting in Figure 1(b) in Gopi et al. (2021), where $p = 0.01$, $\sigma = 0.8$, and $\delta = 0.015$. Middle and Right: The setting of a real application task in federated learning for 10 epochs, with $p = 0.05$, $\sigma = 1$, and $\delta = 10^{-5}$. Here, "EW_EST" is the estimate given by our approximate Edgeworth accountant. We omit the RDP curve in the middle subfigure for better comparisons with others.

**The EEAI.** We now present the empirical performance of EEAI obtained in Section 4.1. We still experiment with NoisySGD. Details of the experiments are in the caption of Figure 3. The two error bounds of EEAI are represented by "EW_UPP" and "EW_LOW", and all other curves are defined the same as in the previous setting. In addition to its optimal time complexity, our analytical finite-sample bounds also achieve better numerical stability for large $m$ in many cases. See Appendix for details.

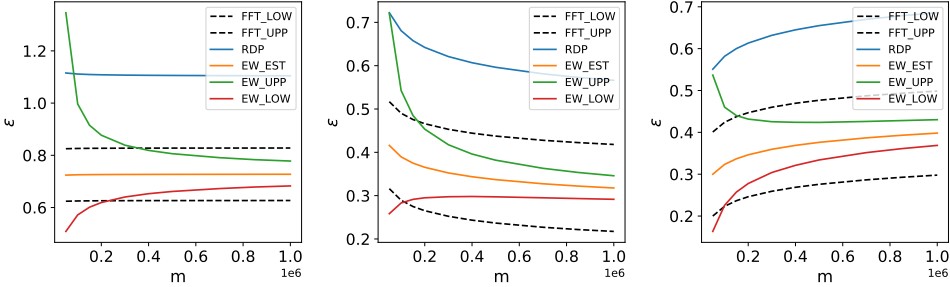

**Figure 3:** We demonstrate the comparisons between our Edgeworth accountant (both AEA and EEAI), the RDP accountant, and the FFT accountant (whose precision of $\varepsilon$ is set to be 0.1). The three settings are set so that the privacy guarantees does not change dramatically as $m$ increases. Specifically, in all three settings, we set $\delta = 0.1$, $\sigma = 0.8$, and $p = 0.4/\sqrt{m}$ (left), $p = 1/\sqrt{m \log m}$ (middle), and $p = 0.1\sqrt{\log m/m}$ (right). We omit the GDP curve here, because the performance is fairly close to the AEA ("EW_EST" curve) when $m$ is large.

## 6 CONCLUSION

In this paper, we provide a novel way to efficiently evaluate the composition of $f$-DP, which serves as a general framework for constructing DP accountants based on approximations to PLLRs. Specifically, we introduced the Edgeworth Accountant, an efficient approach to composing DP algorithms via Edgeworth approximation. In contrast, existing privacy accountant algorithms either fail to provide a finite-sample bound, or only achieve polynomial runtime for general compositions. Importantly, our approach is a complement to the existing literature when the number of compositions is large, which is typical in applications such as large-scale deep learning and federated analytics.

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
