# OpenReview forum: "Analytical Composition of Differential Privacy via the Edgeworth Accountant"
_ICLR.cc/2023/Conference — Submitted to ICLR 2023_

### Official Review · Reviewer_Ri6D · 2022-10-24

**Confidence:** 5
**Clarity, Quality, Novelty And Reproducibility:** 1. Clarity can be improved. The curre…
**Correctness:** 4
**Technical Novelty And Significance:** 2
**Empirical Novelty And Significance:** 1
**Recommendation:** 5

**Strength And Weaknesses:**

Strength: Compared with existing approaches to account for DP guarantee, the proposed method provides a more accurate estimate while incurring lower computation cost. The theoretical analysis seems to be sound.

Weakness:

W1. The trade-off of EA vs. FFT and RDP are not clear.

(1) Regarding the computation cost, how EA and FFT depend on the dimension of the noise is not covered. In particular, in table 1, only m (the number of compositions) is taken into account.

(2) It is not clear why using RDP leads to a loose privacy accountant. Is it because of the conversion rule from RDP to (epsilon, delta)-DP, or some other limitations of RDP? It seems to me that RDP already gives a tight analysis for additive Gaussian noise, since there seems to be no slackness in the derivation in the original paper by Ilya Mironov. For the conversion from RDP to (epsilon, delta)-DP, Canone, Kamath, and Stenike propose an improved conversion rule in "The Discrete Gaussian for Differential Privacy" (see proposition 8); however, this improved rule is overlooked in the current paper.

(3)  Does EA outperforms FFT and RDP when there is no subsampling? If subsampling is not the key factor, perhaps the author could move technicalities about subsampling to the appendix and focus more on EA itself.

(4) Discussion on the effect of order for edgeworth approximation is also missing. How does the order influence the computation and accuracy of the estimate?

W2. Unclear contributions.

It is not clear which propositions/lemmas/theorems in the draft are contributed by the author and which already exist in the literature. Maybe adding references could help.

W3. Issues with experimental settings and results.

(1) The author did not specify the size of the private dataset in their experiments at all. Delta=0.015 is too large for DP applications.

(2) In Figure 3, why do we decrease the sampling probability p while increasing m?

(3) It is unclear why the settings in Figures 2 and 3 are different? It seems that EEAI give the upper and lower bounds for AEA. It is better to put them in the same figures under the same settings.

(4) What is the value of p (for Edgeworth approximation of order p) in the experiments? Is it the same as the subsampling rate p?

(5) Please provide a figure for subsampling rate p=1 in the numerical experiments.

W4. How EA performs in their target applications, noisy-SGD and Federated Analytics, is unclear.

(1) No setting in the experiment section considers noisy-SGD.

(2) The authors only show how their estimate for epsilon is more accurate, but do not provide any result on whether the more accurate estimate translates into better privacy-utility trade-off in the applications: noisy-SGD and Federated Analytics.

W5. Overall, it is questionable if we should choose EA and f-DP to account for the privacy guarantee. As the authors have stated, GDP gives approximately the same result as EA, in the caption of Figure 3. In addition, FFT gives good enough upper and lower bounds for epsilon, as the gains in epsilon seem to be only marginal in Figures 2 and 3. How this more accurate estimate of privacy by EA translates into better privacy-utility trade-off is also not shown in the draft. In addition, the additional computation cost of FFT seems affordable since it is only a one time cost of O(sqrt(m)) and O(m^1.5). Should we choose EA over FFT?

**Summary Of The Paper:**

This paper presents Edgeworth Accountant (EA), an analytical approach to account the privacy loss of differentially private algorithms of multiple iterations (composition). The authors show that EA is more computationally efficient  than the Fast Fourier Transform approach and is more accurate than Renyi-DP accountant.

**Summary Of The Review:**

This paper provides a new tool for the privacy analysis of iterative DP algorithms. The authors claim that their method is more accurate than existing ones, and the computational cost is affordable. The theoretical analysis seems to be sound. However, there are several main concerns about this paper that need to be addressed:

(1) The trade-off of EA v.s. FFT and RDP are not clear

(2) Experimental settings and results

(3) How EA performs in its target applications is not clear

---

> ### Author Response · Authors · 2022-11-12
> **Response to Reviewer Ri6D - Part 1**
>
> We appreciate the reviewer's detailed comments and suggestions on our paper. We response to your questions and concerns as listed in below.
>
> **W1**:
>
> **(1) Computation cost and noise dimension**: Typically, the noise dimension is irrelevant for calculating the composition of $f_i$’s, since it is implicitly accounted for inside the sensitivity analysis. For example, with a subsampled Gaussian mechanism, we add iid noise to each dimension of the parameters, but the $f_i$ is just an 1-dimensional function involving $\sigma$ and $p$, whose evaluation is in $O(1)$ time. This claim is also general, since the DP guarantees (or, specifically $f_i$) for a common mechanism can always be calculated in $O(1)$ time regardless of the dimension of the actual output/noise. This is the reason that we only take $m$ into account for the final time complexity.
>
> **(2) The looseness of RDP**:
>
> That is a nice question. First, as pointed out by the counterexample (page 5 in https://arxiv.org/pdf/2106.08567.pdf), RDP accounting is inherently loose when it converts to $(\varepsilon,\delta)$-DP, i.e., it is impossible to have a perfect conversion rule. This is why RDP can not get a tight $(\varepsilon,\delta)$-DP guarantee even for a simple Gaussian mechanism. Furthermore, things get worse when we move to  the subsampled Gaussian mechanism, where the tight analytical form of RDP is not clear. We note that both factors lead to RDP performing much worse than the other methods (starting from FFT or GDP) in practice, as shown in our paper and many other papers in this field (e.g., Figure 1.b in https://proceedings.neurips.cc/paper/2021/file/6097d8f3714205740f30debe1166744e-Paper.pdf). Back to the reviewer's question, we appreciate the reviewer's pointing out the improved conversion rule and will make changes to our codebase. However, we do not believe it will lead to a huge improvement due to the constraints discussed above.
>
> **(3) Case without subsampling**: When there’s no subsampling, the composition of Gaussian mechanisms admits closed form solution (see Corollary 3.3 in https://arxiv.org/pdf/1905.02383.pdf). Therefore, it is the best case for our EA and RDP, because we provide exact DP guarantees, since the Edgeworth expansion of Gaussian distribution is still Gaussian so the approximation error in this case is zero. It is also true that RDP itself is tight for composition of Gaussian mechanisms, although its conversion to $(\varepsilon,\delta)$-DP will be loose. However, FFT cannot leverage this closed form solution, so it still has some approximation error. Therefore, we believe it makes sense to consider applications of any **approximate accountant** only in the challenging case when closed form guarantee is not possible (i.e. when **subsampling exists**) for a fair comparison among different methods.
>
> **(4) Order of Edgeworth Approximation**: In general, the higher order of Edgeworth expansion, the more accurate the estimate is. Therefore, we believe higher order AEA will give more and more accurate results. Also, the time complexity grows with the order of expansion, since there are simply more terms in the expansion. For EEAI, we spent a lot of effort in getting the finite-sample bounds for the first order Edgeworth expansion, and currently we only implement the first order EEAI. Once we have the exact finite-sample bounds for higher order Edgeworth expansion, we can similarly obtain higher order EEAI. In practice, we found even if we only use first order AEA and EEAI, the privacy guarantee is already accurate when $m$ is large, so we suggest practitioners to use no more than 3rd-order AEA, and use first order EEAI.
>
> **W2: Our contributions**: Thanks. We will make this more clear in the revised version. Specifically, all the theorems and lemmas except Lemma 4.3 are our contribution. We cited the result for Lemma 4.3 just before the statement of the lemma.
>
> **W3:**
>
> **(1) Size of dataset & Delta=0.015**: The overall privacy guarantee is independent of the size of the private dataset, given a fixed subsampling rate $p$. An easy way to see this fact is by noticing the exact characterization of $f_i$ relies on $p$, but not on $n$. Therefore, our experiments only specify $p$, the subsampling rate, for each experiment, and omit the size of the dataset, which is then irrelevant for determining the privacy guarantee. For AEA, we can calculate the approximate DP guarantee for very small $\delta$. The reason that we set $\delta=0.015$ in Figure 2 (Left) is that this is the exact parameter set in the FFT paper by Gopi et al. In Figure 2 (Middle & Right), we set $\delta $ to be $10^{-5}$ which is much smaller, and we still have very good results.

---

> > ### Author Response · Authors · 2022-11-12
> > **Response to Reviewer Ri6D - Part 2**
> >
> > **W3**
> >
> > **(2) decrease $p$ when increasing $m$**: We want to apply EEAI in settings where privacy guarantee is fixed to compare with other methods. And by letting $p^2m$ be a constant preserves privacy guarantee to be asymptotically constant (see for example, equation (5) and the discussion around it in https://arxiv.org/pdf/1911.11607). This is the reason that we decrease the subsampling rate $p$ when increasing $m$. If we keep the same subsampling rate $p$ but increase $m$, the $\epsilon$ will be rapidly growing with $m$, which usually in practice is not allowed. More likely is the case when we have some pre-specified privacy budget on $\epsilon$ and $\delta$.
> >
> > **(3) Different Settings in Figure 2 & 3**: EEAI only provides useful finite-sample bounds when we have very large $m$ (so is the large $m$ in Figure 3), yet AEA provides a very good estimate even for relatively small $m$’s (see Figure 2). In practice, we suggest only using EEAI when $m$ is large.
> >
> > **(4) Using $p$ as Edgeworth order**: Thanks! We will correct this typo in the revision. For EEAI, the order of Edgeworth approximation should be denoted as $k$.
> >
> > **(5) Experiment Setting when p=1**: As we mentioned above, when p=1, there exists a **closed form solution** for the composition of Gaussian mechanisms (see Corollary 3.3 in https://arxiv.org/pdf/1905.02383.pdf), so there is no need to use FFT to calculate the privacy loss approximately. To be specific, for our Edgeworth Accountant with p=1, the PLLRs for Gaussian mechanism have form of $X_i=\log(1-p+pe^{\mu\xi_i-\mu^2/2})=\mu\xi_i-\mu^2/2$, and $Y_i=\log(1-p+pe^{\mu\zeta_i-\mu^2/2})=\mu\zeta_i-\mu^2/2$ (as shown in line 4 of page 6). Here, both $\xi$ and $\zeta$ are Gaussian distributions, and therefore $X$ and $Y$ are sums of Gaussians, which are also Gaussian. Our Edgeworth expansion is exact for Gaussian distribution, which gives 0 approximation error. We feel it is unfair to compare with FFT in the regime where EA gives exact guarantees. (RDP is tight in this case, yet its conversion is still loose.)
> >
> > **W4:**
> >
> > **(1) Noisy-SGD is considered in experiments**: We indeed consider noisy-SGD (and federated analytics) in our experiment, though we do not train a private neural network. The actual training of a neural network is **orthogonal** to calculating the privacy guarantee of the training, and this is why we do not show any training of noisy-SGD, but only provide the privacy guarantee accounting for several cases with different hyperparameters ($p$ and $\sigma$ as in Figure 2 and Figure 3). We believe that this is a tricky point, and we will make sure to emphasize this point more clearly in the revised paper.
> >
> > **(2) privacy-utility trade-off is out of the scope of privacy accounting**: As mentioned above, our experiments track the privacy loss for some noisy-SGD/federated analytics training, without the need to **actually train** those models. The purpose of the current work is to give a better the privacy guarantee for deep learning or federated analytics with large number of compositions. In this way, our works can be viewed as providing a good privacy bound, which is orthogonal to the actual utility of the neural network model. Therefore, it is not feasible to discuss any privacy-utility trade-off in this scope. We note that this is also the case for other existing papers about DP accountant, for example https://arxiv.org/abs/2106.02848.
> >
> > **W5:**
> >
> > **Comparison with GDP**: In fact, as shown in Figure 2, EA has a clear improvement in terms of accuracy when compared to GDP. Also, GDP estimate is only asymptotically correct and it does not come with an exact finite-sample bound.
> >
> > **Comparison with FFT**: When compared with FFT, both our EEAI and FFT provide finite-sample bounds, but our EEAI runs in optimal time complexity. We believe the improvement of time complexity is significant, especially when m is large. The FFT method has time complexity of $O(m^{1.5})$ in heterogeneous case, and $O(m^{0.5})$ in homogeneous case, while our method has time complexity of $O(m)$ and $O(1)$ in those two cases. We have a $O(m^{0.5})$ savings in both cases, which is theoretically important, and practically important when $m$ is very large. Also, our method is numerically more stable than the FFT method when m is large (as demonstrated in Appendix). In practice, whether to choose EA or FFT depends on the specific situations. For compositions with large m, we suggest using EA.
> >
> > **Regarding the privacy-utility tradeoff**: we want to emphasize this is not in the scope of the current paper, as we aim to provide a good DP accountant which is independent of the actual training. A better utility/accuracy can be viewed as a by-product of a better DP accountant, which is similar for EA and FFT when m is large.
> >
> > We would like the reviewer to share any further questions with us. And if we have resolved all your questions, could you please give us more support in your ratings? It really means a lot to us. Thanks!

---

### Official Review · Reviewer_kz4w · 2022-10-24

**Confidence:** 3
**Correctness:** 3
**Technical Novelty And Significance:** 3
**Empirical Novelty And Significance:** 2
**Recommendation:** 3

**Clarity, Quality, Novelty And Reproducibility:**

As mentioned, the paper is unclear about how individual algorithms being composed need to be represented. I guess one can try to trace that from the various bounds and statements, but it would be much more convenient for the reader to provide this information up front.

In terms of novelty, I am not sure how novel two of the main results are: Proposition 3.2 and, especially, Lemma 4.3.

* Proposition 3.2 is similar in spirit to the approach in the Dong, Roth, and Su GDP paper. Is there a new idea here that did not appear in their paper, or is Proposition 3.2 a nice, user-friendly restatement of their approach?

* Lemma 4.3, which is crucial for the "finite sample" bounds, appears identical to Theorem 2.1 in https://arxiv.org/abs/2101.05780. The paper says "we follow the analysis on the finite-sample bound in Derumigny et al. (2021)" which does not clarify whether they are just restating their result (in which case, why is a proof included in the appendix?) or whether they had to modify the Derumigny et al. analysis. If they had to modify it, then why and how?


**Strength And Weaknesses:**

The problem studied by this paper - tight numeric bounds on the privacy parameters of differentially private algorithms - is an important one, and has received a lot of attention recently. The paper gives some evidence that Edgeworth series, applied to the f-differential privacy formalism, give some improvements in terms both of getting tighter bounds and of computational efficiency.

That said, I have the following concerns:
* The paper claims computational complexity improvements, but leaves a lot unclear about this claim.
    * Theoretically, what is the computational model in which the complexity of the algorithms is $O(m)$ for $m$-fold composition? Specifically, what assumptions are made with respect to how individual algorithms $M_i$ being composed are represented? The paper's supplement, when giving pseudocode for the algorithms, says things like "Analytically encode all the corresponding PLLRs". What does this mean? The authors should be precise about what kind of quantities/oracle access to $M_i$ is required, and how these assumptions correspond to the assumptions in prior work.
    * No experimental evidence of better computational efficiency is presented, as far as I can tell. There is some evidence in the supplement of better numerical stability.

* The experimental data presented leaves it unclear how significant the numerical improvements are. The experiments use some arbitrarily chosen (as far as I can tell) values of $\delta$ and present bounds on $\varepsilon$. The values of $\delta$ chosen tend to be rather large: on the order 0.1 and 0.01 for 4 out of 6 plots. The two plots with lower value of $\delta = 10^{-5}$ show that the improvement in the bounds on $\varepsilon$ is rather small compared to the FFT based techniques. Even where the improvement is more significant in relative terms, it is still rather small in terms of the absolute value of $\varepsilon$.

* The paper is not always clear in terms of what is novel. See the next panel.

**Summary Of The Paper:**

The goal of the paper is to present algorithms computing tight non-asymptotic numerical estimates on the privacy parameters of $m$-fold compositions of differentially private algorithms. The paper uses the $f$-differential privacy formalism, which allows cleanly reducing the composition analysis to understanding the tail behavior of sums of independent privacy loss random variables. The original GDP paper of Dong, Roth, and Su used this idea to derive Berry-Esseen type bounds on the privacy loss parameters, as well as asymptotic limit theorems. This paper makes the natural next step (which was also explicitly suggested in the Dong, Roth, Su paper) of using the Edgeworth series approximation of the sum of independent random variables in order to get tighter error bounds.

**Summary Of The Review:**

This may be a nice work, but the write-up leaves a lot unclear about its significance and novelty.

---

> ### Author Response · Authors · 2022-11-12
> **Response to Reviewer kz4w**
>
> We thank the reviewer for the suggestions, and we address all your questions and concerns as listed in below.
>
> **Computational Model**: Thanks for suggesting clarification on the computational model. Here, we assume that we have the oracle of the $f_i$ DP guarantee for each algorithm $M_i$, and we assume the evaluation of any finite moments of PLLRs corresponding to the $f_i$ is in constant time. This is the same model we used for comparing all the methods (esp. FFT). Specifically, with the oracle of $f_i$ (which in our case is the $f_i$ defined by the subsampled Gaussian mechanism), we obtain the analytical form of the PLLRs as specified in Definition 3.1, and we approximate the DP guarantee with the corresponding closed form expression as specified in the Algorithms of AEA and EEAI. Our assumptions are practical in applications, since we can specify the density for the PLLRs, and numerical integration can be used to calculate any moments – crucially, the time of this integration does not depend on $m$, the number of algorithms being composed. Our assumption and computational model is a common practice in studying DP accountant, see also for example the FFT paper by Gopi et al. (https://arxiv.org/abs/2106.02848).
>
> **Efficiency of our methods**: During experiments, we do find several cases when EA outperforms FFT in runtime efficiency, this can be easily seen through the fact that we have a better time complexity than FFT. Given the current theoretical better rate, we affirm to the reviewer that with the increase of $m$, our method will have better efficiency. We can also conduct a simulation for the actual machine time it takes for computing the DP guarantee for FFT and EEAI in the revised paper.
>
> **Significance of our method**: Our method has the optimal time complexity, and when $m$ is large, our method also provides  accurate finite-sample bounds, and is numerically stable being an analytical accountant. We want to clarify that the main focus of our paper is not on the comparison of pure experimental accuracy, but the fact that we provide accurate bounds with optimal time complexity, which is especially important when $m$ is large.
>
> **Novelty of Our Results**
>
> **Proposition 3.2**: Thanks! Proposition 3.2 indeed shares some high-level idea as in Dong et al’s GDP paper, yet is essentially different. The most similar form we found in GDP’s paper is Proposotion 2.12/ Corollary 2.13. Yet, our approach differs from their paper in how we approximate the exact f-DP guarantee (by directly expanding on the summation of random variables), which is also demonstrated in Figure 1. Our approach with Edgeworth expansion provides an exact finite-sample bound, whereas GDP only provides an estimate that is accurate asymptotically. We believe the two methods are different in spirit due to the approximation scheme involved and the finite-sample bound analysis used.
>
> **Lemma 4.3**: We thank the reviewer for mentioning this. Lemma 4.3 is from Derumigny et al. (2021), which we used for our uniform finite-sample bound. We include more proofs and discussions in the appendix to demonstrate how we can potentially **generalize** the bound from first order expansion to higher order expansion. Higher order Edgeworth expansion is not studied in Derumigny et al. (2021) or other literature. We also include the proof for Theorem 1, which is our improvement on Lemma 4.3 leveraging a completely different proof technique to give **adaptive** finite-sample bounds for the first order Edgeworth expansion. This is the first time such a result has been proved in the DP community and the proof technique is new.
>
> We would like the reviewer to share any further questions with us. And if we have resolved all your questions, could you please give us more support in your ratings? It really means a lot to us. Thanks!

---

> > ### Comment · Reviewer_kz4w · 2022-11-14
> > **More about running time**
> >
> > Thank you for your responses. I still have some questions and concerns about running time and the computational model:
> >
> > * You say that you use the $f_i$ curve to "obtain the analytical form of the PLLRs as specified in Definition 3.1", and "approximate the DP guarantee with the corresponding closed form expression as specified in the Algorithms of AEA and EEAI". This is all rather vague -- what is the "analytical form" and what is the "closed form expression" are you talking about, and how are they derived from the $f_i$ curve? It seems you mean computing the PDF or CDF of the privacy loss random variable of the individual mechanisms in a form that allows numerical integration of moments. This is problematic if you want to handle general mechanisms, where there may be no hope for a nice closed form description of the CDF. So, again, I think there is a need here for much more precision. You will notice, for example, that the algorithm descriptions in the Gopi et al. paper specify that they require evaluation oracles for the CDFs of the individual privacy loss random variables, which is a more precise description of the requirements than you give in your paper.
> >
> > * I am still not convinced of the linear running time claim. You say "numerical integration can be used to calculate any moments – crucially, the time of this integration does not depend on $m$, the number of algorithms being composed". This is unclear to me, because numerical integration will only give approximate estimates of the moments, and it's important to know both the convergence rate of the integration method, and the error bound on the moment estimates that your method can tolerate. So, for example, if each moment needs to be estimated to within additive error $O(1/m)$, then the numerical integration step will take time that depends on $m$. As far as I understand, the algorithm in the Gopi et al. paper takes superlinear time because, in order to carry out numerical computations, the random variables need to be discretized to granularity that depend on $m$, and, apriori, it's very possible that something like this will be necessary for your method, too.
> >
> > So, for the running time guarantee to be convincing, I want to see how, given CDFs of the individual PLLRs, you can numerically compute the terms of the Edgeworth expansion, including an analysis of the errors accumulated from things like numerical estimation of moments / cumulants. For a fully general algorithm, this should work with black box access to the CDFs, rather than assuming some analytical representation. Since the significance of your paper hinges on the claimed running time improvements, you need to be precise and rigorous about the computational model and the running time analysis.

---

> > > ### Author Response · Authors · 2022-11-17
> > > **Thanks!**
> > >
> > > Thanks for bringing up the point on the form of $f_i$ as well as a more detailed description on the computation model (especially numerical integration). We address these two points separately as follows.
> > >
> > > **Closed form expression**: Regarding computing the moments of a privacy loss random variable, we would like to clarify that each $f_i$ corresponds to a simple mechanism like a subsampled Gaussian mechanism (like one single iteration in DP-SGD). For general mechanisms, we believe each individual $f_i$ of this general composed mechanism is still simple, or at least we require it to be a white-box model, where we know **what noise is added**, and therefore the PLLRs of this $f_i$ can be analytically written as a function of the density of this noise. To be specific, PLLR is the log likelihood ratio of a single (subsampled) mechanism under $P_i$ and $Q_i$ specified by $f_i$. We demonstrate how to calculate this for Gaussian and Laplacian mechanisms, and the only thing we require in those calculations is simply the density of those added noise. We feel this should be a mild requirement on the mechanism, and as the reviewer points out, we need some oracle: we can either directly have the oracle of PDF/CDF of the PLLRs, or we can compute those PLLRs from the oracle of PDF of the added noise. Compared to the result in Gopi et al., they also require the oracle of CDF of a single PRV for each mechanism before composition. In practical settings such as sub-sampled Gaussian mechanisms, analytical PDF exists and can be numerically evaluated efficiently.
> > >
> > > **Discretization error for numerical integration**: Thanks! We remark that our original claim ignores most of the numerical errors, but focus more on the actual theoretical error by the exact Edgeworth Accountant. But, even when we account for the numerical error, the time complexity is also much better than the FFT based method, which we will explain as follows. To be more precise, the reason we omit this kind of numerical error is that even storing the number $m$ itself requires $O(\log m)$ space and time complexity. If we account for this, then our time complexity for **homogeneous composition is $O(\log m) = \tilde{O}(1)$**, and **heterogeneous composition is $O(m\log m) = \tilde{O}(m)$**, where $\tilde{O}$ are omitting logarithmic terms of $m$.
> > >
> > > The reason for our better time complexity when we account for numerical integration error is because we only need to calculate (upto 4-th) moment of the PLLRs, and sum them together to calculate $G_{m, k, X}$ and $\Delta_{m, k, X}$. When $m = 1$, if we require any moment’s discretization error to be $\tau$, let’s say the number of partitions in the discretization to achieve this requirement is at least $n_1$. Then when $m > 1$, the worst case discretization error of the summation of the moment becomes $O(m \tau)$.  So if we want to control the entire discretization error of the sum of all moments to be still $\tau$, each integration needs to have a discretization error of $\tau’ = \frac{\tau}{m}$, which requires the partition to be of order $o(n \log m)$. To see this, we can do integration using any standard or reasonable way like Gaussian quadrature, and with $n$ partition in the discretization, the error is bounded by a rate of $O(n^{-n})$ (Here we start from the classic error estimate of form $\frac{(b - a)^{2n + 1}(n!)^4}{(2n + 1) (2n!)^3}f^{(2n)}(\eta)$, and assume the density of PLLR is smooth enough, so the higher order derivative is not growing faster than some $c^n$ for some constant $c > 1$. This condition on $f$ can be verified for common mechanisms like Gaussian or Laplacian). Therefore, to achieve the same discretization error for $m > 1$ and $m= 1$, we need a $o(\log m)$ multiplicative overhead to our computation time of each numerical integration. However, the time to express $m$ is even longer than this overhead, therefore, we need a $O(\log m)$ term for both the heterogeneous and homogeneous compositions, and therefore, we will have time complexity of $O(\log m)$ and $O(m\log m)$ for homogeneous and heterogeneous composition.
> > >
> > > I hope those resolves your concerns, and we really appreciate your questions! We will make all these points more clear in the revised paper to be more precise.

---

> > > > ### Comment · Reviewer_kz4w · 2022-11-21
> > > > **IMO the paper needs a major revision**
> > > >
> > > > Thank you for your clarifications. I think your paper needs to be carefully re-written to clarify these issues and flesh out the computational model and numerical error analysis you sketch in your response. As you revise your paper, I encourage you to avoid using vague terminology like "analytically written" when discussing your algorithms, and instead precisely state the oracle access needed, and the assumptions you need to make. It is clear from your responses that your results need a number of assumptions (e.g., that PLLR density is sufficiently smooth) that your current write up does not mention. If these issues are clarified, and if you add formal error analysis for the numerical integration steps, then I imagine the rewritten paper will be an interesting addition to the differential privacy literature. As currently written, however, I don't think the paper is ready for publication.

---

> > > > > ### Author Response · Authors · 2022-11-25
> > > > > **Thanks**
> > > > >
> > > > > We thank the reviewer for acknowledging that our paper will be an interesting addition to the current privacy literature. We believe our paper is solid in the proof, novel in the analysis, and is an important complement to the existing privacy accountant methods. We feel the numerical integration error is an interesting perspective that we can discuss more, but is a minor issue we already resolved in the above discussion. Note that our main focus is on the privacy accountant with better time complexity, and the computational complexity is significantly better than the FFT based method even if we would like to account for the numerical integration error. Importantly, our time complexity analysis still holds true if the cumulants can be calculated precisely, and our main massage is the same for general cases since we reduce the time complexity from $O(\sqrt{m})$ to $\tilde{O}(1)$ in homogeneous case, and from $O(m^{1.5})$ to $\tilde{O}(m)$ in heterogeneous case. Also, we believe the additional discuss of numerical error is simple enough, and since it is resolved in the above discussion, we can make this minor change easily in the following camera ready revision.
> > > > >
> > > > > Again, we appreciate the reviewers valuable suggestion in discussing the numerical issue, and we will definitely want to update this discussion in the revision. Also, I believe the above discussion indeed resolve the reviewer's concern about the assumptions & correctness of the time complexity. Therefore, even it is a revision, it will be a minor one with just a few additional analysis solely on numerical integration which only involves a specific implementation. Therefore, we sincerely hope the reviewer could only make this a minor revision, and we can update it in the camera ready version. Thank you a lot!

---

### Official Review · Reviewer_iZYg · 2022-10-25

**Confidence:** 3
**Correctness:** 4
**Technical Novelty And Significance:** 4
**Empirical Novelty And Significance:** 4
**Recommendation:** 6

**Clarity, Quality, Novelty And Reproducibility:**

The quality, novelty, and reproducibility of the paper are all high. As mentioned before, the ideas in the paper are all "natural in retrospect" but at the same time a great deal of technical effort is needed for the main results in the paper. Also, the Edgeworth accountant obtains several novel guarantees both qualitatively and quantitatively improving over past work.

As mentioned before, the authors do a good job explaining the concepts in the paper such that the main ideas are "natural in retrospect", so in that sense the paper is mostly clear. However, to someone without the appropriate background reading through the technical discussions that solidify these ideas may be difficult.

**Strength And Weaknesses:**

The paper has two main strengths. First, it gives a very tight and efficient accounting scheme, that qualitative and quantitatively improves on past accounting schemes heavily. Accounting schemes are widespread in practice and there is good reason to believe the improvements on accounting will allow us to give stronger DP guarantees in practice (or equivalent, achieve much better accuracy for a target DP guarantee). The qualitative and improvements of the accountant are also good reason to hope it becomes widely used in practice. Second, the ideas leading up to the Edgeworth accountant are slick and feel "natural in retrospect", but the paper nonetheless is technically very novel, and the proofs are involved and several complex lemmas are needed to turn the ideas into an algorithm.

The main weakness of the paper is that a lot of technical background is assumed or quickly skipped over in the exposition. One would imagine the ideal reader for this paper is someone who e.g. maintains an actively used privacy accounting library and is interested in working on implementations of the Edgeworth accountant. It may be hard for such a person to read even through the body of the paper, since I don't believe the Edgeworth expansion is common knowledge among privacy researchers (while something like f-DP might be). Admittedly, this is probably at least somewhat true of most sufficiently technical papers given the page limit. But if it is possible to include even slightly more background on the Edgeworth expansion, it would make the paper much more approachable to the audience of interest. Of course, this may be difficult due to space constraints.

Nitpick: I think Proposition 3.2 or some part of its buildup should explicitly state/remind that f_i is the tradeoff given by thresholding the PLLRs, right now this needs to be inferred.

**Summary Of The Paper:**

The paper introduces the Edgeworth accountant, an accountant for composing DP mechanisms that obtains tighter guarantees and/or is faster to evaluate than existing accountants. The Edgeworth accountant of the paper is the first accountant to provide both finite-sample (i.e., non asymptotic in m) lower and upper bounds, with O(1) computational complexity for composing m iid mechanisms and O(m) for m non-iid mechanisms (here I believe the O() hides some dependences on the order of the Edgeworth approximation, though empirically the authors show a small order suffices). In contrast, the FFT accountant of Gopi et al. takes time O(sqrt(m)) and O(m^5/2) respectively, the moments accountant only gives upper bounds for non-asymptotic computations, and the Edgeworth refinement to the GDP accountant of Zheng et al. does not give a finite-sample guarantee.

The Edgeworth accountant (presented in the body of the paper for composing subsampled Gaussians, but which the authors show can be extended to other noise distributions common in the DP literature such as the Laplace mechanism) uses two ideas. First, it uses the fact that the optimal hypothesis test for which of two databases a DP output came from can be defined only in terms of the output's privacy-loss log-likelihood ratio (PLLR). In turn, the authors can show that composition of the f-DP guarantees given by the PLLR hypothesis tests, is equivalent to an (eps, delta)-DP guarantee written strictly in terms of the CDFs of the sum of the PLLRs. Second, these CDFs can be approximated using the Edgeworth expansion, a generalization of the central limit theorem which approximates the CDF of a distribution in terms of its cumulants. So, one can also approximate the (eps, delta)-DP guarantee given by the first idea, using the Edgeworth expansion. In order to obtain a strict upper/lower bound on the actual DP guarantees, one also needs a bound on the CDF approximation error given by the Edgeworth expansion, which the authors derive, and tighten for the special case of subsampled Gaussians.

The authors implement the (second-order) Edgeworth mechanism and show that in various settings, (i) the approximation given by their accountant is sandwiched between the upper and lower bounds given by the FFT accountant of Gopi et al, and much more accurate than RDP/CLT-based accountants, and (ii) the interval on epsilon given by their accountant is contained strictly within the interval given by the FFT accountant, and RDP is far outside both these intervals.

**Summary Of The Review:**

Overall I think the paper is above the acceptance threshold. As mentioned before, I think the paper is strong theoretically, contains some nice new ideas, and has a lot of potential for practical impact. That being said, it is not clear to me that the presentation of the paper, especially to practitioners who may lack some of the statistical background, is as accessible as it could be.

---

> ### Author Response · Authors · 2022-11-12
> **Response to Reviewer iZYg**
>
> We would like to thank the reviewer for mentioning the two strengths of our paper: We provide an efficient and tight DP account with optimal time complexity; and we also provide solid analysis on the DP bound and some of our proof techniques are also new.
>
> We also thank the reviewer for the comment on the writing style on technical background. Indeed, we defer some background of Edgeworth expansion to the appendix due to the page limit since we want to convey all our main message within the page limit. We totally understand the need to make it more approachable for the general audience with limited background in Edgeworth expansion. We will try to do that in the revised paper. In the camera-ready version we will try our best to include more background on the Edgeworth expansion. Edgeworth expansion is just like a higher order Taylor expansion to CLT, we will give more references.
>
> For Proposition 3.2 we will explicitly remind the reader again about the definition of $f_i$ in revision, thanks for the suggestion.
>
> We would like the reviewer to share any further questions with us. And if we have resolved all your questions, could you please give us more support in your ratings? It really means a lot to us. Thanks!

---

### Official Review · Reviewer_ESEo · 2022-11-01

**Confidence:** 2
**Correctness:** 3
**Technical Novelty And Significance:** 3
**Empirical Novelty And Significance:** 3
**Recommendation:** 6

**Clarity, Quality, Novelty And Reproducibility:**

The paper is well written, quality is high. I did not verify the reproducibility of the paper.

**Strength And Weaknesses:**

The fact that they can compute the privacy cost for composition is constant for identical privacy mechanism and just linear in the general case. The paper also gives numerical experiments to substantiate their claim.

I really did not see much of a weakness in the paper. I really liked reading the paper and if I am not missing something, this paper has definite improvement over previous work. The paper is well written. I have not yet verified the proof, and that is the only reason I am not giving full support in accepting the paper.

**Summary Of The Paper:**

The paper gives computationally efficient privacy bounds for the composition of DP algorithm with finite sample guarantees. The algorithm runs in constant time to compute the privacy loss for m identical DP mechanism, and in general case, the run time in $O(m)$ time. This improves on the previous results that use FFT. The proposed approach in this paper is Edgeworth accountancy.

**Summary Of The Review:**

Please see above.

For correctness below, I have not read the proofs so I cannot make a judgement on the correctness of the paper; however, none of the claims seems to be out of ordinary. Hence a rating of 3. Once I have verified the proofs, I will move it to 4.

---

> ### Author Response · Authors · 2022-11-12
> **Response to Reviewer ESEo**
>
> We thank the reviewer for acknowledging our strengths and the good quality of the paper. We would like to ask the reviewer to give more affirming support for the paper when you have time to check our proof. We really appreciate your help and it means a lot to us. In fact, the tools we used in the proof are all well-known probabilistic tools, though the idea is novel. And we are confident our result is correct and sound. Please let us know if we can help explaining anything.

---

### Decision · Program_Chairs · 2023-01-20

**Decision:**

Reject

**Justification For Why Not Higher Score:**

Overall no enthusiasm and some serious concerns from an expert.

**Justification For Why Not Lower Score:**

N/A

**Metareview: Summary, Strengths And Weaknesses:**

This paper presents a new accounting method for DP composition. Accurate composition accounting is central for DP analysis of ML training methods and this paper presents a more computationally efficient approach way to perform accounting. This is a promising work, but, as pointed out in the review of kz4w, the computational efficiency and underlying assumptions are not discussed with sufficient detail and clarity for the paper to be accepted at this stage.